# HARP: Unsupervised Histopathological Artifact Restoration

**Moritz Fuchs**[1]                                            MORITZ.FUCHS@GRIS.TU-DARMSTADT.DE
[1] *Technical University Darmstadt*

**Ssharvien Kumar**[1]                     SSHARVIEN.KUMAR.SIVAKUMAR@GRIS.TU-DARMSTADT.DE
**Mirko Schöber**[1]                                   MIRKO.SCHÖBER@GRIS.TU-DARMSTADT.DE
**Niklas Woltering**[2]                                  WOLTERING@MED.UNI-FRANKFURT.DE
[2] *Johann Wolfgang Goethe University Medical School*

**Marie-Lisa Eich**[3]                                       MARIE-LISA.EICH@CHARITE.DE
[3] *Universitätsmedizin Berlin - Charité*

**Leonille Schweizer**[2,3]                       LEONILLE.SCHWEIZER@MED.UNI-FRANKFURT.DE
**Anirban Mukhopadhyay**[1]                   ANIRBAN.MUKHOPADHYAY@GRIS.TU-DARMSTADT.DE

## Abstract

Histopathological analysis, vital for medical diagnostics, is often challenged by artifacts in sample preparation and imaging, such as staining inconsistencies and physical obstructions. Addressing this, our work introduces a novel, fully unsupervised histopathological artifact restoration pipeline (HARP). HARP integrates artifact detection, localization, and restoration into one pipeline. The first step to make artifact restoration applicable is an analysis of anomaly detection algorithms. Then, HARP leverages the power of unsupervised segmentation techniques to propose localizations for potential artifacts, for which we select the best localization based on our novel inpainting denoising diffusion model. Finally, HARP employs an inpainting model for artifact restoration while conditioning it on the artifact localizations. We evaluate the artifact detection quality along with the image reconstruction quality, surpassing the state-of-the-art artifact restoration. Furthermore, we demonstrate that HARP improves the robustness and reliability of downstream models and show that pathologists can not tell the difference between clean images and images restored through HARP. This demonstrates that HARP significantly improves image quality and diagnostic reliability, enhancing histopathological examination accuracy for AI systems.

**Keywords:** Histopathology, Artifact Restoration, Diffusion Models, Unsupervised

## 1. Introduction

Histopathological analysis stands at the forefront of diagnostic medicine, informing critical decisions with life-altering implications. Yet, the reliability of such analysis is often compromised by artifacts introduced during sample preparation and imaging, ranging from staining inconsistencies (Tellez et al., 2019) to physical obstructions like folds and blood cells (Kanwal et al., 2022). These artifacts can distort the data, leading to diagnostic inaccuracies of the employed AI (Schömig-Markiefka et al., 2021; Wang et al., 2021b).

While in clinical practice, whole slide images (WSI) can be rescanned to address these issues, the restoration of corrupted histopathological images with an AI model offers an effective alternative to improve image quality without such **labor and time-consuming** process. Recent methods have ventured into this territory with supervision-heavy approaches,

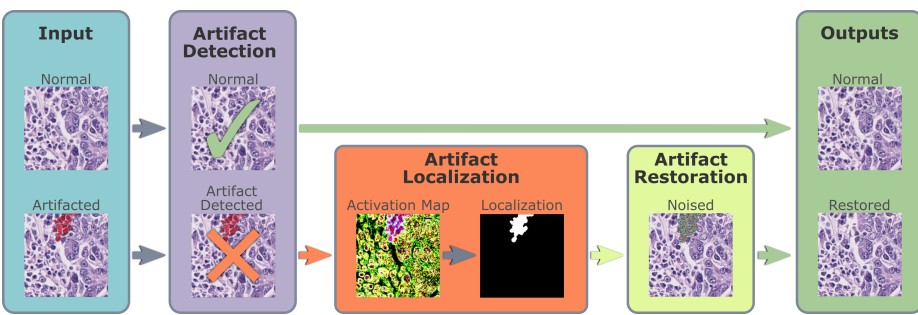

Figure 1: **Overview of Histological Artifact Restoration Pipeline (HARP)**

demanding extensive manual input (He et al., 2023) or supervision (Dahan et al., 2022; Ke et al., 2023) on patches containing artifacts in order to restore a WSI. Approaches that explicitly learn each artifact type's appearance by supervision will struggle with unseen artifacts and fail to restore the image (He et al., 2023), especially where unseen artifacts are common and manual intervention is costly. These supervisions usually render such approaches unreliable and labor-intensive when pursuing a holistic pipeline. To the best of our knowledge, there has not been a holistic, unsupervised approach that detects the artifacts and performs image restoration in one pipeline. Recognizing this gap, we introduce a fully unsupervised Histopathological artifact restoration pipeline (HARP) that deploys the three steps depicted in Figure 1, which are essential for a clinical workflow for computational pathology: **artifact detection**, **artifact localization**, and **artifact restoration**.

In order to make HARP useable in the clinical workflow, the first step is to reliably detect artifacts. Many studies in recent years have developed unsupervised anomaly detection methods, which are able to identify unusual images (Wang et al., 2021a; Yu et al., 2021; Zavrtanik et al., 2021). We evaluate anomaly detection methods with the AnomaLib framework (Akcay et al., 2022) on histopathology images with realistic and proven synthetic artifacts from Stieber et al. (2022). In the second step, we require the localization of the relevant artifacts. This step is crucial for the applicability of our pipeline to generalize to many different artifact types without requiring knowledge of them. Based on the original image, we generated multiple localization masks of artifacts by leveraging pre-trained knowledge of SAM (Kirillov et al., 2023) and clustering with DBSCAN (Ester et al., 1996). As we train a diffusion model to restore the images, we leverage it to generate an activation map, which we deploy to rank the top 5 masks. In the final step, we conduct the artifact restoration. We generate a restored image for each localization by conditioning our diffusion model to inpaint the image based on the localization mask. Recent works (He et al., 2023) have shown great results on artifact inpainting with the RePaint (Lugmayr et al., 2022) and manually annotated artifacts. This approach has the limitation of being computationally heavy, rendering it unsuitable for a clinical workflow. HARP leverages our novel inpainting denoising diffusion model, incorporating the condition input at every step, which reduces the required computational time. Lastly, we select the final image with the artifact detection method. In summary, our contributions are threefold. We evaluated existing anomaly detection methods for histopathological artifacts. Secondly, we develop a **(I) novel conditional image inpainting denoising diffusion model**. Further, we

demonstrate its capability for **(II) artifact localization and restoration**, and we evaluate the **(III) impact of HARP on downstream model performance**. Lastly, we evaluate the restored images by conducting a **(III) study with pathologists**.

## 2. Related Work

In the niche of histology imaging, there have only recently been efforts to restore artifacts, in contrast to the extensive work in general computer vision where object removal and image inpainting is a long-standing topic (Zhang et al., 2023). In standard computer vision applications, techniques for object removal, such as RePaint (Lugmayr et al., 2022), Palette (Saharia et al., 2022), DDRM (Kawar et al., 2022), or GAN-based networks (Zhang et al., 2020), have been employed to address issues like shadow elimination and image blurring. These methods, however, typically hinge on certain prerequisites: target predictability, pre-defined object masks, clean image pairs, or extensive datasets (Xiang et al., 2023).

Despite these advancements, these methods have limited direct application for histopathological image restoration, and data scarcity and annotation costs are rampant issues in medical imaging. In recent years, many papers have tackled the topic of stain normalization (Faryna et al., 2021; Wagner et al., 2022) and data augmentation (Tellez et al., 2019) in order to improve model generalization. However, the domain of artifact restoration in histopathological images has only seen the emergence of the first supervised methodologies(Dahan et al., 2022; Ke et al., 2023; He et al., 2023), requiring considerable manual input or supervision on artifacts for effective WSI restoration. While these approaches represent innovative strides, they are often impractical for real-world application due to their inability to cope with unseen artifacts or require costly manual input. Consequently, their assessments lack real clinical settings complexities and variability. These inherent limitations undermine the reliability and are impractical in clinical practice. HARP addresses this gap by proposing an approach that focuses on preserving critical features, thereby ensuring that the artifacts are restored without compromising the fidelity of the histological images.

## 3. Method

HARP involves three steps that we are explaining in order. First, we give a brief explanation of the **artifact detection**. Then, we explain the training of the novel inpainting diffusion model, which we leverage for **artifact localization** and **artifact restoration**.

**Artifacts Detection:** In HARP, the initial step is the efficient and reliable detection of artifacts, which is crucial for effective restoration in clinical workflows. Utilizing AnomaLib, we explored various anomaly detection methods tailored for histopathology images. Among these, we identified FastFlow (Yu et al., 2021), which employs a Vision Transformer (ViT) Encoder to conduct normalizing flow on latent representations, as the most suitable for our needs. This method demonstrated superior performance in artifact detection and was easily integrated into our pipeline. The implementation of this method marks the beginning of our artifact detection phase in HARP, as illustrated in Figure 2, setting the stage for accurate histopathological image restoration and enhancing diagnostic reliability.

**Training the Conditional Diffusion Model:** The purpose of conditional generative models is to estimate the data distribution $p(y|x)$, where $x \in [0, 1]^{H \times W}$ is a conditional

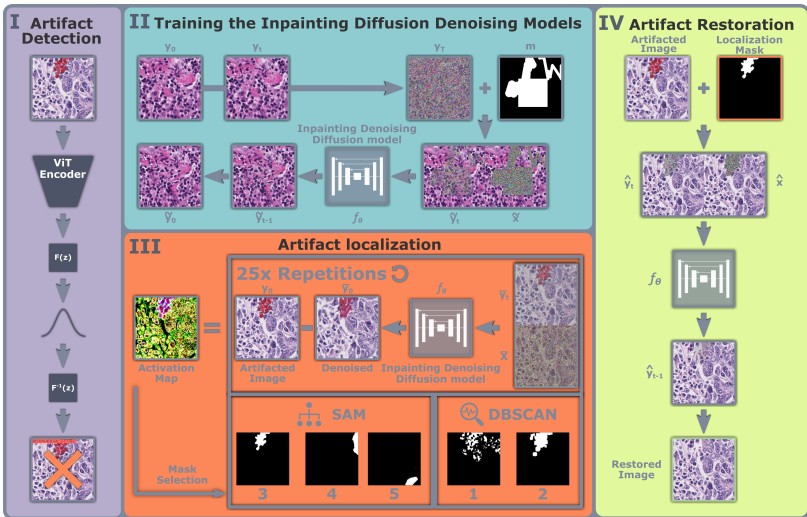

Figure 2: **Detailed overview of the methods deployed in HARP:** (I) Detecting artifacts with FastFlow. (II) Our training process of the inpainting diffusion denoising model $f_\theta$, with synthetic localization masks. (III) Artifact localization pipeline, for which $f_\theta$, together with SAM and DBSCAN, selects 5 localization masks. (IV) Artifact restoration inference with image and localization masks. The best restoration is selected by the artifact detection.

image and $y \in [0,1]^{H \times W}$ is the target, with height $H$ and width $W$. Given this goal, Denoising Diffusion Models (DDMs) operate by reversing the process of gradually introducing Gaussian noise into image samples $y_0 \sim p(y_0)$. After a series of $T = 1000$ diffusion steps, the resulting sequence $y_1, ..., y_T$ converges towards a Gaussian noise profile, particularly as $T$ approaches infinity. Given a well-calibrated variance schedule $\beta_1, ..., \beta_T \in (0,1)^T$, small steps and large $T$, we train a denoising model $f_\theta$ to reverse each step in this sequence. The reverse distribution is defined by the conditional: $p_\theta(y_{t-1}|y_t, x) := \mathcal{N}(y_{t-1}; \mu_\theta(x, y_t, t), \Sigma_\theta(x, y_t, t))$.

Here, $\mu_\theta$ and $\Sigma_\theta$ are predictions given the model parameters $\theta$. To condition our denoising model $f_\theta$ for inpainting, we randomly mask out an area $\tilde{x} := y_T * m + y_0 * (1 - m)$, with a binary mask image $m \in \{0,1\}^{H \times W}$. With $p(y_T|\tilde{x})$ being a known distribution, one can initiate the reverse process by sampling $y_T \sim \mathcal{N}(0, \mathbf{I})$ and iteratively applying the denoising model, thereby transforming it back into the conditional data distribution. The training of $f_\theta$ is simplified to the following loss (Ho et al., 2020):

$$\mathbb{E}_{y_0 \sim p(y_0), \epsilon \sim \mathcal{N}(0,1), m, t}[\| f_\theta(\tilde{x}, \underbrace{(\sqrt{\bar{\alpha}_t} y_0 + \sqrt{1 - \bar{\alpha}_t} \epsilon) * m + y_0 * (1 - m)}_{\tilde{y}_t}, t) - \epsilon \|_1] \qquad (1)$$

with $\epsilon \sim \mathcal{N}(\mathbf{0}, \mathbf{I})$, and the factorization $\alpha_t := 1 - \beta_t$ and $\bar{\alpha}_t := \prod_{s=1}^t \alpha_s$. The novel approach of incorporating $\tilde{x}$ in each step of the denoising process is critical for our model's performance and allows for fewer RePaint (Lugmayr et al., 2022) cycles during artifact restoration.

**Artifact Localization:** In order to localize potential artifacts without supervision, we gather an activation map of our **novel denoising diffusion model** $f_\theta$ when conditioned on the entire noised image $\bar{x} := \bar{y}_T$. If we noise an artifact image $\bar{y}$ up to 900 steps and denoise again, all the major features tend to stay the same, but minor details on known structures change slightly. As our model does not know artifacts, minor details do not change on artifacts. Using this, we calculate the reconstruction error: $\|\bar{y}_0 - y_0\|$, which aggregate as an activation map over 25 denoising repetitions. We prompt SAM (Kirillov et al., 2023) and DBSCAN (Ester et al., 1996) to generate object localizations on the artifact image $y_0$. We remove masks that are too large($> 60\%$) and small($< 0.4\%$), as reconstruction would be unfeasible or not worthwhile, and sort out duplicated masks. Then, we score each mask by aggregating the activations over the mask area and select the top 5 binary masks $m_0, ..., m_4$ based on the lowest activation received. Finally, we dilate the masks to smooth the inpainting boundary of artifacts. Most importantly, this is - to the best of our knowledge - the first method to **localize histological artifacts without any supervision**.

**Image Restoration:** In order to restore an artifact image $\hat{\tilde{y}}_0$, we condition on an image $\hat{x} := \hat{y}_T$ with $\hat{y}_t := \hat{\tilde{y}}_t * \hat{m} + \hat{y}_0 * (1 - \hat{m})$, where $\hat{m}$ is one of the previously determined artifact localizations. To harmonize the masked area with the image, we apply a denoising procedure to our novel inpainting denoising diffusion model similar to RePaint (Lugmayr et al., 2022) with $resampling = 3$ and $jumpsampling = 10$ to generate the restored and artifact-free image $\tilde{y}$. The best reconstruction from the 5 gets chosen by FastFlow (Yu et al., 2021), having the least probability of containing an artifact.

## 4. Experiments and Results

In this section, we start by introducing the dataset, artifacts, and backbones. Second, we evaluate artifact detection methods for each artifact. Next, we quantify the restoration qualities of our model with ground truth artifact masks and our localization maps. The most important ability for artifact restoration is its application in the clinical workflow for computational pathology. For this reason, we evaluate a state-of-the-art downstream model on artifact images with and without the use of HARP and conduct a user study to determine whether there is a difference between a clean image and the outputs of HARP.

**Dataset and Artifacts:** We evaluate all methods of the Breast Cancer Semantic Segmentation dataset (BCSS) (Amgad et al., 2019), for which we use an FCN8 architecture proposed in the paper for the downstream task evaluation. While BCSS contains a multitude of labels, we focus on the four predominant classes (% of labels): tumor (45%), stroma (17%), lymphocyte-rich tissue (35%) and necrosis (3%). We train our inpainting denoising diffusion model and the downstream task model using 11075 training, 1031 validation and withhold 1000 test patches. The diffusion model backbone is based on the guided diffusion model from github.com/Janspiry/Palette-Image-to-Image-Diffusion-Models. From each region of interest within a WSI, we sample random crops of size 600x600 with 0.24 mpp, which we resample to 256x256. We train our models using an NVIDIA RTX 4090. Our code is available here: github.com/MECLabTUDA/HARP . Leveraging previous works (Stieber et al., 2022), we adopted the following artifacts: **dark spots**, **fat drop**, **squamous epithelia**, **threads**, **blood cell** and **blood group**, **compression**, **cuts**, **overlap** and **folding**,

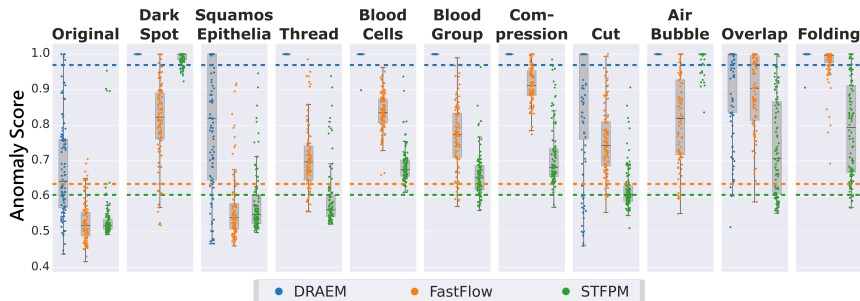

Figure 3: **Evaluation of Artifact Detection per Artifact Type**

which have been shown to be realistic and detrimental to downstream performance by (Wang et al., 2021b; Babendererde et al., 2023). We generate 100 samples for each artifact.

**Artifact Detection:** The first step to deploying artifact restoration efficiently in a clinical workflow is to detect artifacts reliably. We evaluate various methods from Anoma-Lib (Akcay et al., 2022) on the 4 local artifacts from Schömig-Markiefka et al. (2021), which can be found in the appendix. From these methods, we chose the best three methods: DRÆM (Zavrtanik et al., 2021), FastFlow (Yu et al., 2021) and STFPM (Wang et al., 2021a) and validate them using the validation set to determine the quantile for which each method keeps 95% of normal samples (dashed lines). Figure 3 shows how this affects the detection of all artifact types. We select the FastFlow method for the pipeline, as it only misses most artifacts from squamos epithelia and has the best accuracy with 85.8%. DRÆM and STFPM are strong competitors; DRÆM misses most of the artifacts for squamous epithelia, cut, and overlap. DRÆM average accuracy is 83.9%. STFPM, on the other hand, misses squamos epithelia, thread, and cut artifacts with an average accuracy of 74.9%.

**Artifact Restoration Quality:** Now, we want to assess the image quality generated by the artifact restoration model and the full HARP pipeline. As HARP is, to the best of our knowledge, the first fully unsupervised method, we compare to three methods that either **require artifact images during training** like AR-CycleGAN (Ke et al., 2023) or **manual input by providing localization masks** like ArtiFusion (He et al., 2023) and DDRM (Kawar et al., 2022). To provide localizations reliably to all models, we leverage the ground truth segmentation masks of the artifacts from Stieber et al. (2022). We train ArtiFusion and DDRM on the BCSS with the same backbone architecture as our model.

Table 1: Image quality metrics for the artifact restoration of the first five artifacts.

| Artifact: | Dark Spot | | Squamos Epi. | | Thread | | Blood Cells | | Blood Group | |
|---|---|---|---|---|---|---|---|---|---|---|
| Metric: | FID↓ | PSNR↑ | FID↓ | PSNR↑ | FID↓ | PSNR↑ | FID↓ | PSNR↑ | FID↓ | PSNR↑ |
| Supervised methods with Ground Truth Masks | | | | | | | | | | |
| **ArtiFusion** | 78.3 | 19.6 | 25.7 | **24.4** | 40.4 | 21.3 | 48.3 | 22.7 | 54.1 | 20.7 |
| **AR-CycleGAN** | 98.4 | 17.0 | 77.8 | 19.4 | 114.2 | 17.7 | 179.3 | 17.3 | 130.6 | 16.9 |
| **DDRM** | 85.8 | **20.6** | 38.1 | 24.2 | 51.5 | **21.4** | 65.6 | 21.8 | 61.9 | 20.8 |
| **Ours** | **70.3** | 18.9 | **25.2** | 24.1 | **34.2** | 21.2 | **33.8** | **23.7** | **40.1** | **21.1** |
| Unsupervised method without Ground Truth Masks | | | | | | | | | | |
| **HARP (Ours)** | **64.3** | 17.2 | 50.2 | 20.6 | 44.7 | 19.4 | 192.0 | 16.3 | 86.4 | 16.8 |

Table 2: Image quality metrics for the artifact restoration of the last five artifacts.

| Artifact: | Compression | | Cut | | Air Bubble | | Overlap | | Folding | |
|---|---|---|---|---|---|---|---|---|---|---|
| Metric: | FID↓ | PSNR↑ | FID↓ | PSNR↑ | FID↓ | PSNR↑ | FID↓ | PSNR↑ | FID↓ | PSNR↑ |
| Supervised methods with Ground Truth Masks | | | | | | | | | | |
| **ArtiFusion** | 44.5 | 22.2 | 46.1 | 21.9 | 54.9 | 19.0 | 29.5 | 23.2 | 41.3 | **20.8** |
| **AR-CycleGAN** | 129.9 | 18.4 | 124.6 | 17.5 | 145.8 | 17.5 | 86.4 | 17.9 | 125.9 | 16.8 |
| **DDRM** | 52.8 | **22.5** | 53.2 | **22.3** | 69.1 | **19.4** | 41.0 | 23.1 | 50.8 | **20.9** |
| **Ours** | **38.1** | 22.4 | **43.9** | 22.0 | **44.9** | 18.7 | **26.7** | **23.5** | **35.4** | 20.4 |
| Unsupervised method without Ground Truth Masks | | | | | | | | | | |
| **HARP (Ours)** | 65.3 | 17.9 | 60.5 | 19.0 | 66.3 | 17.1 | 51.6 | 18.0 | 69.6 | 17.4 |

Further, as **AR-CycleGAN requires a dataset representing artifacts**, we used the original dataset and deployed it to our test cases. We evaluate 100 artifact images per artifact type in Table 1 and 2 for Fréchet Inception Distance (FID ↓) and Peak Signal Noise Ratio (PSNR ↑), where the arrow indicates the direction of improvement. Our artifact restoration model performs best for **13 out of 20 results**, and DDRM offers minimal improvements on 5 of the metrics and ArtiFusion on 1. Ours has a **reduced runtime of 18.6 sec.** per image vs. 30.9 sec. for ArtiFusion and 37.4 sec. for DDRM. AR-CycleGAN fails to generalize to the unseen artifact domains, demonstrating supervised training limits. When looking at our fully unsupervised HARP method, we see that it can even improve on the dark spot artifact, which is likely due to a better mask. Evaluating the artifact localizations and the ground truth masks by DICE, HARP achieves 54.5%. However, HARP limitations are when the localizations are suboptimal, e.g., for blood cells and groups. This is likely due to one of three reasons: First, the artifact detection failed. Second, these artifacts are often spread out over the whole image, contrary to other artifacts. Third, the training set likely contains some artifacts, leading to a reproduction of the same artifacts in the image. However, when the segmentation mask is appropriate, the results are excellent, as supervised results show. The key area for improvement lies in the quality of unsupervised localization masks, as they significantly contribute to the artifact restoration process.

**Downstream Application:** To evaluate the usability of HARP for the clinical workflow of computational pathology, we evaluate the segmentation performance of the downstream model using clean images, artifact images, artifact images excluding artifact segmentations, and restored images with HARP. These images all have the same underlying image and segmentation mask from the test, for which we calculate the DICE score per class and the average. It is important to note that AI-generated images pose a risk for accurate diagnoses, similar to undisclosed deepfakes; therefore, we ensure transparency in our process by excluding these contents using HARP's artifact localization. As seen in Table 4, the performance of the artifact images significantly decreases compared to the originally clean images. HARP is able to recover the artifact images and effectively reduces the performance drop introduced by the artifacts by 48%, which makes the downstream model more robust and reliable for the daily clinical workflow.

**User Study:** Finally, we conducted a user study with four pathologists on 50 image pairs as a visual turning test on the produced image quality. One of the images from the pair is a normal image from the training distribution, and the other is an image from the test distribution augmented with an artifact and then processed with HARP. We use 5

Table 3: Downstream performance of state-of-the-art segmentation model on BCSS for clean, artifact and images restored with HARP.

| Metric: | DICE % | | | | |
|---|---|---|---|---|---|
| on: | Tumor | Stroma | Lymphocyte-rich | Necrosis | Average |
| **Clean** | $86.1 \pm 0.4$ | $83.8 \pm 0.7$ | $81.8 \pm 2.1$ | $74.2 \pm 2.8$ | $81.5 \pm 1.1$ |
| **Artifacts** | $77.7 \pm 0.3$ | $77.9 \pm 0.9$ | $76.2 \pm 2.3$ | $64.9 \pm 2.6$ | $74.2 \pm 1.0$ |
| **Artifacts wo. seg** | $80.6 \pm 0.4$ | $81.1 \pm 0.8$ | $77.9 \pm 2.4$ | $68.6 \pm 2.9$ | $77.0 \pm 1.1$ |
| **HARP (Ours)** | $82.2 \pm 0.5$ | $82.0 \pm 0.9$ | $78.5 \pm 2.3$ | $69.3 \pm 2.8$ | $78.0 \pm 1.0$ |

artifact images from each of the 10 artifact types. The pathologists were given instruction to conduct the study on 256x256 images with 100% scale and not to zoom to avoid image interpolation artifacts from the preprocessing of all images affecting the study. The study was timed in order to ensure that participants followed a standard clinical workflow. We calculate the Matthews correlation coefficient (MCC) for each participant and give the number of falsely classified samples. The participants achieved the following scores: -0.071 MCC (27/50), -0.159 MCC (30/50), 0.239 MCC (20/50), and 0.296 MCC (18/50) with the times 7:34 min, 5:50 min, 4:45 min, and 7:00 min, respectively. All our participants found it impossible to tell the real difference between images, as our results suggest, at best, there is a weak positive correlation by chance. This further demonstrates the potential of HARP and the risks of not disclosing generated content for the clinical workflow. We give more results and a sample of five images from the study in the appendix.

**Dangers and Impact:** Generative AI can not go unlabeled – as the EU AI Act (EU, 2021) suggests. In pathology, Generative AI risks misleading the diagnosis done by pathologists and other AIs, which we showed with our user study. Therefore, we exclude the artifact localization masks in the downstream evaluation and recommend highlighting them. Nonetheless, computational pathology has the promise of saving time and increasing diagnostic accuracy for patients, for which HARP is a supportive structure to ensure reliability. Further, it has the dual benefit of improving the image quality without rescanning.

## 5. Conclusion

In conclusion, our work presents the **Histopathological Artifact Restoration Pipeline (HARP)**, a novel and fully unsupervised approach – **the first of its kind**– for restoring artifacts in histopathological images. HARP efficiently integrates artifact detection, localization, and restoration into one seamless clinical workflow for computational pathology. Our work significantly enhances image quality and diagnostic accuracy by leveraging advanced **artifact detection**, deploying a **novel unsupervised artifact localization** technique, and presenting a new **state-of-the-art inpainting denoising diffusion model**. We evaluate the artifact detection quality along with the image reconstruction quality, surpassing the state-of-the-art artifact restoration methods. Furthermore, we demonstrate that HARP can improve the robustness and reliability of downstream models. Finally, we show that **pathologists can not tell the difference** between clean images and images processed through HARP. This breakthrough in medical image processing holds immense potential for improving AI systems' reliability in histopathological examinations.

## Acknowledgments

This work was supported by the Bundesministerium für Gesundheit (BMG), Germany with grant [ZMVI1-2520DAT03A] and by the Bundesministerium für Bildung und Forschung (BMBF) with grant [01KD2210B]. Special thanks to Simon Streit for his valuable contribution to this study.

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

## Appendix A. AnomaLib preliminary Evaluation

Table 4: AnomaLib preliminary method evaluation: This table shows the attempt to compare anomaly detection algorithms included in the Anomalib framework. The results show how well the trained algorithms perform on a test dataset augmented with the four local artifacts from (Schömig-Markiefka et al., 2021). The algorithms are compared using the metrics AUROC, F1-Score, and accuracy. Algorithms marked with * were trained on a smaller dataset of 650 images due to hardware limitations. Algorithms marked with ** have been limited to 30 training epochs due to time constraints. The experiment was run three times with different random seeds. We select the three highlighted methods (DRÆM, FastFlow, and STFPM) for further evaluation.

| Method | AUROC ↑ | F1-Score ↑ | Accuracy ↑ |
|---|---|---|---|
| CFA | $0.742 \pm 0.031$ | $0.689 \pm 0.016$ | $0.648 \pm 0.026$ |
| CFLOW | $0.702 \pm 0.022$ | $0.671 \pm 0.018$ | $0.573 \pm 0.029$ |
| DFM | $0.655 \pm 0.014$ | $0.743 \pm 0.008$ | $0.656 \pm 0.014$ |
| **DRÆM \*\*** | $\mathbf{0.872 \pm 0.015}$ | $\mathbf{0.819 \pm 0.009}$ | $\mathbf{0.823 \pm 0.010}$ |
| Efficient AD \*\* | $0.627 \pm 0.020$ | $0.667 \pm 0.001$ | $0.501 \pm 0.002$ |
| **FastFlow \*\*** | $\mathbf{0.869 \pm 0.025}$ | $\mathbf{0.784 \pm 0.020}$ | $\mathbf{0.777 \pm 0.021}$ |
| PaDiM \* | $0.683 \pm 0.017$ | $0.695 \pm 0.004$ | $0.563 \pm 0.009$ |
| PatchCore \* | $0.683 \pm 0.002$ | $0.699 \pm 0.000$ | $0.569 \pm 0.000$ |
| Reverse Distillation | $0.489 \pm 0.059$ | $0.663 \pm 0.004$ | $0.496 \pm 0.004$ |
| **STFPM \*\*** | $\mathbf{0.769 \pm 0.033}$ | $\mathbf{0.688 \pm 0.014}$ | $\mathbf{0.624 \pm 0.085}$ |

## Appendix B. Artifact Localization Evaluation

Table 5: DICE scores between the ground truth (GT) mask from (Stieber et al. 2022) and our unsupervised localizations for each type of artifact. We further compare HARP localizations **undilated (u.)** with the GT masks to assess whether our initial localization under-segments the GT; as a result, we see that the dilation in HARP is necessary. The DICE only decreases significantly for dark spots. However, we see in Table 1 that this has a positive impact on the image quality.

| Artifact: | Dark Spot | Squamos Epithelia | Thread | Blood Cells | Blood Group | Com-press-ion | Cut | Air Bubble | Over-lap | Fold-ing |
|---|---|---|---|---|---|---|---|---|---|---|
| Metric: | DICE [%] | | | | | | | | | |
| **GT vs. (Ours) HARP** | 66.6 | **31.1** | **83.8** | **20.9** | **50.9** | **60.4** | **42.9** | 68.1 | **58.7** | **62.3** |
| **GT vs. HARP (u.)** | **68.4** | 30.0 | 80.7 | 18.4 | 48.7 | 60.2 | 40.5 | **68.2** | 57.8 | 61.0 |

## Appendix C. Qualitative Examples

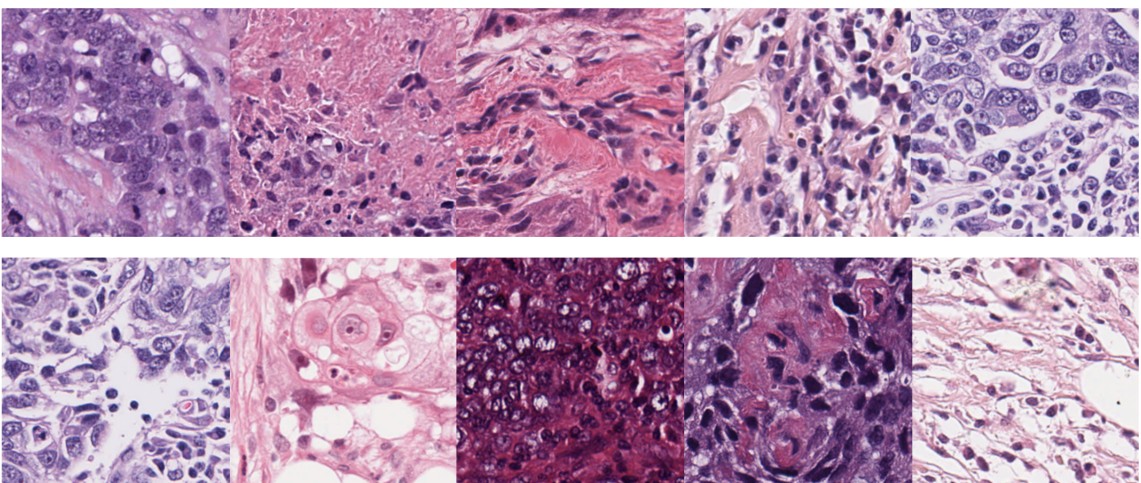

Figure 4: Qualitative samples from the study: We used pairs of images (top and bottom), where one is from the original training distribution and one is a test sample that had an artifact, which got removed from HARP. We then asked the participants to label which image had an artifact and went through the pipeline.
Spoiler: Right answers from left to right:
Bottom, Top, Top, Bottom, Top

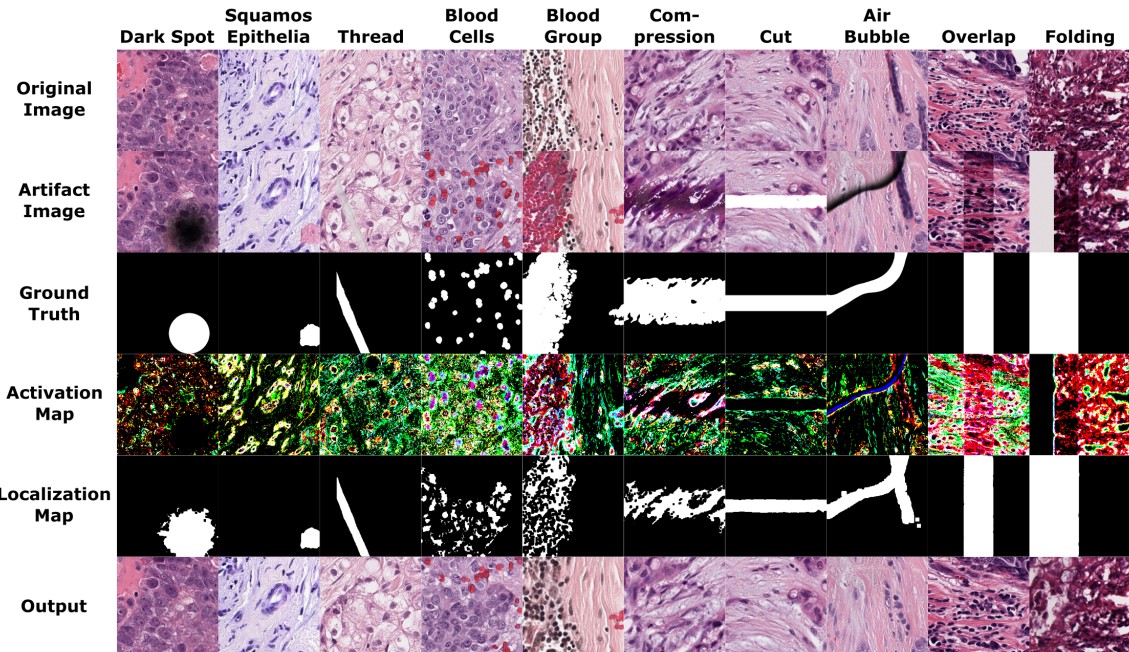

Figure 5: Qualitative samples from HARP for each artifact: We selected random samples from the HARP. For the blood cell and group artifacts, it clearly shows the failure case, when not everything is selected by the localization mask, usually due to a lack of a better localization proposal. The Air Bubble case is a rare sample that has a natural fold/compression artifact that got segmented with the air bubble and removed.

## Appendix D. Video of HARP in Action

Load HARP Video                                    Click the symbol

Figure 6: Illustrative movie of the whole HARP Pipeline. Playing works with Adobe Acrobat.

