# OpenReview forum: "HARP: Unsupervised Histopathology Artifact Restoration"
_MIDL.io/2024/Conference — MIDL 2024 Poster_

### Official Review · Reviewer_jxzF · 2024-02-28

**Confidence:** 3
**Preliminary Rating:** 3
**Recommendation:** Poster
**Final Rating:** 3.5

**Summary:**

This paper proposes a series of unsupervised methods for detecting and removing artifacts in histopathology images.

Pipeline: Anomaly detection using FastFlow --> artifact localization using the difference from the original vs corrupted-and-then-denoised output, followed by segmentation of the localization heatmap --> image restoration using the artifact image masked with the predicted segmentation.

Results are compared against baselines using image quality metrics and downstream segmentation performance. The restored images are visually compared against clean images by pathologists.

**Strengths:**

The method can be completely unsupervised with no annotation during test-time.

Numerous state-of-the-art methods are put together to obtain reasonable performance boost from existing solutions.

Thorough evaluations are performed for each step of the pipeline.

**Weaknesses:**

In Tables 1 & 2, HARP seems to perform worse than baselines for most artifact types. The best performance is achieved by "Ours", which if I understood correctly, uses the ground-truth segmentation and then performs the rest of the restoration. This seems to speak more about the effectiveness of the diffusion model for restoration rather than the unsupervised aspect of HARP.

The novelty seems to be mostly in the application of a collection of existing methods rather than a new method. FastFlow, RePaint, SAM/DBSCAN are all existing methods. The claimed novelty on the "novel diffusion model" is unclear -- there's only one sentence mentioning "incorporating $\tilde{x}$ in each step of the denoising process". This sounds like a simple input channel concatenation.

Some explanations should be improved. Details are in the comments below.

**Detailed Comments:**

It is puzzling how, even though HARP did not seem to perform very well in terms of quantitative metrics, the user study seems to indicate that pathologists could not tell the difference between clean vs restored images. Is there a good explanation for this phenomenon? Are the metrics suboptimal for determining the quality of restoration? Were the average metrics skewed due to a few badly restored images?

How are the masks generated for training the diffusion model?

Labels are missing for Fig. 2-III, the two inputs into the diffusion model. It's hard to decipher from the text and the rest of the figure what those images are supposed to represent (e.g. is one of them supposed to be $\bar{x}:=\bar{y}_T$? Wouldn't that be just random noise?).

The rationale behind using the reconstruction error as "activation map" is unclear. Is this from an empirical observation? Am I understanding correctly that artifacts will have lower error? Why is reconstruction better for artifacts if the reconstruction is performed to inpaint real images? Also, if other studies have used similar approaches, there should be a citation.

"we ensure transparency in our process by excluding these contents using HARP’s artifact localization" - what are the "contents" being excluded in this sentence?

**Justification Of Final Rating:**

Thank you to the authors for the detailed responses.
It seems that some of the responses have been added to the manuscript, but some have been left as responses here in the review comments. Many of the questions arose from incomplete or hard-to-find info from the manuscript, so I believe it could have been better for these comments to be reflected in the manuscript itself.
I still feel a bit reserved about the concatenation of input alone being a noteworthy novelty, but I understand the argument that the combination of all other components can be an important contribution as well. Given the responses here and the amount of changes in the manuscript, I will slightly increase my rating to borderline accept.

**Justification Of The Preliminary Rating:**

The study seems to be thorough and combines a lot of interesting methods. However, I have a few concerns regarding the novelty and usefulness of the unsupervised method, especially given some discrepancies in the quantitative evaluation metrics and visual inspections by pathologists. I also feel that some explanations for the core principles are missing (e.g. why was the activation map chosen that way? what exactly is novel about the diffusion model? etc.).

**Questions To Address In The Rebuttal:**

Any responses to the weaknesses and detailed comments would be good for the rebuttal.

**Special Issue:**

No

---

> ### Author Response · Authors · 2024-03-08
> **Answers to jxzF**
>
> # Weaknesses:
> ### “In Tables 1 & 2, HARP seems to perform worse than baselines for most artifact types. The best performance is achieved by "Ours", which if I understood correctly, uses the ground-truth segmentation and then performs the rest of the restoration. This seems to speak more about the effectiveness of the diffusion model for restoration rather than the unsupervised aspect of HARP.”
>
> As there exists no fair comparison to HARP, we opted to compare all on fair grounds by giving them the ground truth segmentation masks. In this case, our method -- the Artifact restoration model of HARP --  clearly outperformed all the comparisons resulting in  new SOTA, while being computationally more efficient. In a real world application, no supervised method would get a ground truth segmentation mask to base their inpainting on. As there are no comprehensive datasets that include a high diversity of annotated natural artifacts, we decided to make our model HARP fully unsupervised. The results of this unsupervised approach should be seen as something with no direct comparison as other method like ArtiFusion would require to inpainting the full image, which is clearly a loss of valuable that can be reliably examined without the potential of misclassification based on some generative AI.  As we found the only two reproducible methods that produced somewhat decent results from the histopathology domain, we adapted DDRM to our problem to improve the number of comparisons. We think this field clearly deserves more attention and further research, that’s why we make our code accessible after the acceptance.
> ### “The novelty seems to be mostly in the application of a collection of existing methods rather than a new method. FastFlow, RePaint, SAM/DBSCAN are all existing methods. The claimed novelty on the "novel diffusion model" is unclear -- there's only one sentence mentioning "incorporating in each step of the denoising process". This sounds like a simple input channel concatenation.”
>
> Thank you for your valuable feedback on our manuscript. I would like to address the concerns raised regarding the novelty. There has **not** been an **unsupervised artifact localization method in histopathology**, yet. Further, **our diffusion model integrates the addition of $\tilde{x}$** to be concatenated to the input, which allows us to **reduce time-consuming procedures like RePaint and gain improved results**. This last novelty alone would be enough improvement in itself so that other people would justify a standalone paper. Further, our method also outperforms all other SOTA supervised methods. However, we do not like publishing minimal publishable novelties. We add **our novel artifact localization** and artifact detection to evaluate the first **truly unsupervised approach for artifact restoration**. That's why it might seem like a lot is going on that has been already done, which is only to show the full potential of your novel approach.

---

> ### Author Response · Authors · 2024-03-08
> **Answer to jxzF comments**
>
> # Comments:
> ### “It is puzzling how, even though HARP did not seem to perform very well in terms of quantitative metrics, the user study seems to indicate that pathologists could not tell the difference between clean vs restored images. Is there a good explanation for this phenomenon? Are the metrics suboptimal for determining the quality of restoration? Were the average metrics skewed due to a few badly restored images?”
> We agree this is a problem of the metrics images that did not detect, or artifacts were only partially inpainted received awful scores, severely impacting the average. This can be seen especially for blood-related artifacts. For the user study, we only selected images that had artifacts detected and therefore were inpainted. However, missing minor parts of the artifacts, e.g. for blood cells, might have gone unnoticed by the clinicians as the augmentation looks quite realistic. This also shows a limitation of the study, clinicians are trained to classify/diagnose patients and ignore the artifacts, which might have caused them to miss such instances. However, some likely showed a minor correlation because of such leftover artifacts.
> ### “How are the masks generated for training the diffusion model?”
> Thank you for the excellent question. It is a combination of diagonal mask, box mask, free-form stroke mask and mask covering the entire image. 10% of the time diagonal mask that covers have the image from random corner. 10% of the time, a box mask that covers a random corner of the image. 10% of the time mask that cover the entire image. 70% of the time, union of box mask at a random spot of the image and free-form stroke mask from. The free-form Image inpainting uses gated convolution. Mask covering the entire image is to make sure the diffusion model still learn the entire image, which is required as we wanted to use the same diffusion model to do reconstruction of the entire image for artifact localization. The entire implementation will be available in a GitHub repository upon acceptance.
> ### “Labels are missing for Fig. 2-III, the two inputs into the diffusion model. It's hard to decipher from the text and the rest of the figure what those images are supposed to represent (e.g. is one of them supposed to be $\bar{x}:=\bar{y}_T$? Wouldn't that be just random noise?).”
> Thank you for this comment; we improved the figure to make this clear. To your question: yes, that would be just random noise, as we have no prior on the location of the artifacts. One could potentially integrate a region proposal here to improve the activation map and therefore the selection of potential artifact localizations, as we anticipated this change for the future we opted to use the more abstract form of writing, but made it now more explicit.
> ### “The rationale behind using the reconstruction error as "activation map" is unclear. Is this from an empirical observation? Am I understanding correctly that artifacts will have lower error? Why is reconstruction better for artifacts if the reconstruction is performed to inpaint real images? Also, if other studies have used similar approaches, there should be a citation.”
> We agree that the reconstruction map is not a real activation map as known from CAM or similar methods. However, the activation map encodes how much the diffusion model could accurately reproduce-- what is known to the model– that's why we see it as an activation map over time in the diffusion process. This is also why artifacts usually have a lower score in reconstruction error, as the model does not know how to alter such content, they are mostly not represented in the training data. For known contents, like cells and stroma etc., the diffusion model builds a more robust representation of them from the training, making the diffusion model to propose just very slight alterations. In order to increase this effect, we aggregate the activation map over 25 passes to improve the consistency. Concept-wise, this is close to anomaly detection methods like FastFlow and Draem; however, they do not use diffusion models and therefore do not have the time to unroll the denoising objective. If there are better citations that are closer to what we are doing, please bring them to our attention.
> ### “"we ensure transparency in our process by excluding these contents using HARP’s artifact localization" - what are the "contents" being excluded in this sentence?”
> We exclude the inpainted areas that we determined with our artifact localization. In order to assess the value of the artifact restoration, we added results only excluding these areas without restoring them, and we see that the artifact restoration has a significant effect on the downstream model performance. We hope this helped to clarify your question.
>
> Thank you very much for your detailed review, we hope we could clear up all of your questions. Feel free to ask additional questions when we left something untouched.

---

### Official Review · Reviewer_yaJT · 2024-03-02

**Confidence:** 3
**Preliminary Rating:** 4
**Recommendation:** Poster
**Final Rating:** 4

**Summary:**

this is a paper about histopathology image artifact restoration. Unsupervised learning methods are used for artifact localization first. Then novel diffusion based inpainting method is used for restoration. Good (SOTA) results were obtained in the experiments. Authors showed that the proposed model, HARP, also improves robustness and reliability of downstream tasks too. Pathologists were not able to tell the difference between clean and restored images.

**Strengths:**

- the problem is not addressed before with the way authors proposed.
- there are merits in the work due to use of inpainting, diffusion, and experimental setup where both quantitive markers and then pathologists' remarks (qualitative) compared.
- downstream tasks were also improved with robustness and reliability.
-motivation of the work is given well. Paper is written in a nice flow.

**Weaknesses:**

- inpainting with diffusion is not totally new as it has been already used in Computer Vision applications, perhaps incrementally new in medical imaging.
- colors in the figures are making it hard to understand and depict what is going on.
- too much bold faces make the reading difficult.
- comparisons can be strengthened but I dont see this as a major problem.

**Detailed Comments:**

my both weakness and strengths comments are self-contained, please see them.

**Justification Of Final Rating:**

paper has a good rebuttal period, many questions were answered positively, figures were not informative and authors promised to improve them, not sure how to check them before giving final score,

overall,  since it is not super innovative, I think it is a weak accept, fair conference paper.

**Justification Of The Preliminary Rating:**

The paper is presenting incrementally novel idea of diffusion based inpainting and new application of artifact removal from histopathological images. results look promising, downstratem tasks are at good performance, and the paper is overall written nice. Comparisons are at place, and this is a fair conference paper at MIDL, I believe.

**Questions To Address In The Rebuttal:**

- how does this inpainting affect the quantification ? are those region represented in the final quantitative results in clinical value? or they are skipped due to artifacts? we can't rely on those regions, right?

**Special Issue:**

No

---

> ### Author Response · Authors · 2024-03-08
> **Answers to yaJT**
>
> # Weaknesses:
> ### “inpainting with diffusion is not totally new as it has been already used in Computer Vision applications, perhaps incrementally new in medical imaging.”
> Certainly, the use of inpainting with diffusion models is a well-established technique in the field of Computer Vision. The application in medical imaging has it own intricacies and specific requirements to potentially offer new challenges and opportunities for this technology. However, we do propose novelties that have shown specific application to medical images. Therefore, the impact to pathology could be substantial and certainly worthy of exploration and publication.
> ### “colors in the figures are making it hard to understand and depict what is going on.” and “too much bold faces make the reading difficult.”
> Thank you for your feedback, we do like these colors as they are easy to recognize and associate with the different parts of HARP. However, we try to adapt the figures more easily readable, including a high dpi in order to allow for zooming in on close for better reading. Further, we revised the text to contain significantly less bold faces in order to make reading easier. We hope you like the changes.
> ### “comparisons can be strengthened but I dont see this as a major problem.” and “how does this inpainting affect the quantification ? are those region represented in the final quantitative results in clinical value? or they are skipped due to artifacts? we can't rely on those regions, right?”
> As requested by other reviewers, we do include the performance of the downstream task only excluding the artifact localizations without restoring them in order to better assess the value of artifact restoration alone. Further, we added the variance to the downstream task in Table 3. These results show that the artifact restoration improves the downstream model even though all correspondingly inpainted pixel are excluded in the evaluation.
>
> Generative AI should never go unlabeled – as the EU AI Act suggests. Especially in pathology, Generative AI poses the risks of misleading the diagnosis done by pathologists and other AI systems, which we showed with our user study. That’s why we exclude the artifact localization masks from the downstream evaluation and would recommend highlighting them with a transparent overlay when presented to the pathologist in order to make sure they see which parts have been altered by an AI. In case of doubt, a pathologist should switch to the original image taken by the scanner or even look at the raw sample. Nonetheless, computational pathology has the promise of saving time and increasing diagnostic accuracy for patients, for which HARP could be a supportive structure to ensure the reliability of such an AI system. Further, it has the dual benefit of improving the image quality for the pathologist viewing an image without the need to rescan/remount the sample. We included a shorter version of this paragraph as discussion of this problem.  However, we encourage future research into this topic in order to improve the reliability of restoration, more efficient and broadly applicable tools in order to achieve a trustworthy AI that revolutionizes the daily clinical routine and that’s why we make our code available upon acceptance.
>
> Thank you very much for your review, we are very delighted about your feedback and interesting questions.

---

> > ### Comment · Reviewer_yaJT · 2024-03-23
> >
> > thanks for the responses

---

### Official Review · Reviewer_DNG7 · 2024-03-04

**Confidence:** 5
**Preliminary Rating:** 2
**Recommendation:** Poster

**Summary:**

The paper presents a pipeline for unsupervised artefacts detection in histopathology images, including artefact localisation and restoration with pinpointing.

**Strengths:**

The methodology is well-explained and motivated. It also includes a comparison of several methods for artefacts detection based on unsupervised out-of-distribution detection. Evaluation of the artefact restoration process on downstream tasks is also included.

**Weaknesses:**

The pipeline is a combination of several existing methods, without much methodological novelty. My main concern is the artefact restoration process, which in my opinion is unnecessary and constitutes solving a more difficult task (restoring the image/eliminating the artefacts) than needed. As histopathology images are vast, an area with an artefact once detected can simply be ignored/not processed in downstream processing. The process of restoration is dangerous as it can introduce non-existent information or remove important information (in the case of of a false positive).

**Detailed Comments:**

Confidence intervals should be included in Table 3, otherwise it is not possible to assess the performance of the methods relative to each other.

A true baseline comparison would be a method that ignores artefact areas in a downstream tasks.

While I appreciated the informal tone of the caption of Figure 4, perhaps it is not very suitable for a scientific papers and the authors should reconsider.

**Justification Of The Preliminary Rating:**

The paper has limited methodological novelty. I find the advocated use of restoration for artefacts in histopathology problematic (particularly without discussion of limitations and dangers) and perhaps unnecessary.

**Questions To Address In The Rebuttal:**

The dangers and limitations of the "restoration" component of this pipeline should be discussed.

A better baseline should be presented that compares the methods to simply ignoring detected areas with artefacts.

---

> ### Author Response · Authors · 2024-03-08
> **Answers to DNG7**
>
> # Weaknesses
> ### “The pipeline is a combination of several existing methods, without much methodological novelty.”
> Thank you for your valuable feedback on our manuscript. I would like to address the concerns raised regarding the novelty. There has not been an unsupervised artifact localization method in histopathology, especially  one that leverages diffusion models with pre-trained knowledge of a foundation model. Further, our diffusion model integrates the addition of x tilde to be concatenated to the input, which allows us to reduce time-consuming procedures like RePaint and gain improved results. Our diffusion model outperforms all SOTA supervised methods. Further, we add our novel artifact localization and artifact detection to evaluate the first truly unsupervised approach for artifact restoration.
> ### “My main concern is the artefact restoration process, which in my opinion is unnecessary and constitutes solving a more difficult task (restoring the image/eliminating the artefacts) than needed. As histopathology images are vast, an area with an artefact once detected can simply be ignored/not processed in downstream processing.”
> While we agree that artifact restoration is a more difficult task to do, but it is necessary to improve the overall viewing experience of the pathologist.  The alternative of rescanning/remounting the sample is too time-consuming to be a viable option for clinical pathologists under time pressure. We also added results for the downstream task when we do not inpaint the artifact localization and just exclude it from evaluation, and the restoration has a clear positive impact on the downstream model, further establishing the benefits.
> # Comments
> ### “Confidence intervals should be included in Table 3, otherwise it is not possible to assess the performance of the methods relative to each other. A true baseline comparison would be a method that ignores artefact areas in a downstream tasks.”
> Thank you, that's an excellent idea, we added variances and the results by just ignoring the artifact localization of the artifact image in table 3.
> ### “While I appreciated the informal tone of the caption of Figure 4, perhaps it is not very suitable for a scientific papers and the authors should reconsider.”
> Thanks, we removed the “encouragement to try it out yourself”. We will include the full study in the GitHub repository up on acceptance.
> # Questions
> ### “The dangers and limitations of the "restoration" component of this pipeline should be discussed.” and also the weakness: “The process of restoration is dangerous as it can introduce non-existent information or remove important information (in the case of of a false positive).”
> Generative AI should never go unlabeled – as the EU AI Act suggests. Especially in pathology, Generative AI poses the risks of misleading the diagnosis done by pathologists and other AI systems, which we showed with our user study. That’s why we exclude the artifact localization masks from the downstream evaluation and would recommend highlighting them with a transparent overlay when presented to the pathologist in order to make sure they see which parts have been altered by an AI. In case of doubt, a pathologist should switch to the original image taken by the scanner or even look at the raw sample. Nonetheless, computational pathology has the promise of saving time and increasing diagnostic accuracy for patients, for which HARP could be a supportive structure to ensure the reliability of such an AI system. Further, it has the dual benefit of improving the image quality for the pathologist viewing an image without the need to rescan/remount the sample. We included a shorter version of this paragraph as discussion of this problem.  However, we encourage future research into this topic in order to improve the reliability of restoration, more efficient and broadly applicable tools in order to achieve a trustworthy AI that revolutionizes the daily clinical routine and that’s why we make our code available upon acceptance.
> ### “A better baseline should be presented that compares the methods to simply ignoring detected areas with artefacts.”
> We added the results in Table 3 of only excluding the artifact localization from evaluation and not applying the artifact restoration. This shows that the artifact restoration still comes with an added benefit for the downstream model. Further, it has the dual benefit of improving the image quality for the pathologist viewing an image without the need to rescan/remount the sample. However, this restoration should be made transparent to the pathologists, as discussed before.
>
> Thank you very much for your review, we are very delighted about your feedback and interesting questions.

---

> > ### Comment · Reviewer_DNG7 · 2024-03-27
> >
> > I would like to thank the authors for addressing some of my comments and remarks. It is not clear what the added "confidence intervals" are, I assume it is mean +- std? If that is the case, I would argue that there is no substantial difference between processing the slides with the detected artefacts excluded and inpainting the artefacts with the proposed method.
> >
> > According to the authors using the inpaintin has the additional benefit of making the viewing more pleasant for pathologists, but I would argue that pathologists would prefer to see the original image, and certainly not an inpainted image with the inpainted regions annotated as suggested. This would be an unnecessary complications, and a potentially dangerous one.
> >
> > Because of this I will keep my original assessment unchanged.

---

> > > ### Author Response · Authors · 2024-03-27
> > > **Answer to Reviewer DNG7 Comment**
> > >
> > > Thank you for your input. The confidence intervals indeed represent the mean ± standard deviation. We've validated the improvement from inpainting over just excluding artifacts localizations of HARP with a significant two-sided t-test (p=4.89e-9).
> > >
> > > The remaining concerns seem to stem primarily about difference in opinions. We believe we have adequately addressed these within the limitations section of our paper and do not foresee additional revisions in this area.
> > >
> > > We appreciate your engagement and are open to further discussion.

---

### Official Review · Reviewer_ukLv · 2024-03-05

**Confidence:** 4
**Preliminary Rating:** 4
**Recommendation:** Poster
**Final Rating:** 5

**Summary:**

In this paper, the authors proposed HARP, an unsupervised histopathological artifact restoration pipeline. This framework detects, localizes and restores the image with artifacts. With ground truth masks, their method outperforms the SOTA methods. Although in a fully unsupervised manner, it still fall behind current supervised methods, it is very interesting and useful in practice.

**Strengths:**

Conducting artifact removal in an unsupervised manner is very interesting and useful. As mentioned in the paper, it avoids the tedious labeling process and it is a valuable area to explore. The authors claimed that this is the first unsupervised method for artifact removal in pathology.

**Weaknesses:**

To my understanding, the performance of the proposed method heavily depends on the quality of the 1st step (artifact detection) and 3rd step (artifact localization).
1. In the 1st step, the authors used a general anomaly detection method for artifact detection. It is a general method, I have a concern about it performance in the pathology domain.
2. Some parts in the 3rd step are unclear to me.

**Detailed Comments:**

Major concerns:
For artifact detection:
1. The authors used a general anomaly detection method, which was not trained on pathology images. The authors showed some comparison among general anomaly detection methods but how does it actually perform compared with supervised methods in digital pathology? Anomaly detection itself is a very difficult problem and I have a concern on whether it can generalize well in the pathology domain.
2. Will it generate many false positive region on images without artifacts? As most patches in WSIs do not contain artifacts, even if its possibility is not large, it may have a large influence. I also wondered how it affects the downstream tasks.

For artifact localization:
1. The authors used activation maps from the diffusion model to localize artifacts. But it is unclear to me why it works and how the diffusion model helps in this stage.
2. "As our model does not know artifacts, minor details do not change on artifacts." As the authors mentioned later, "the training set likely contains some artifacts". As we know, most WSI datasets contain artifacts, how does this affect the performance of the proposed methods? It may have a large influence if the scale the dataset to a much larger size.

Minor ones:
1. What's the prompt to the SAM, masks or dots? If a patch contains several disjoint artifacts, do they prompt together or separately?
2. What's the y axis in the figure 3? I also suggest adding some captions for this figure. It is not clear now.

**Justification Of Final Rating:**

The authors proposed a pipeline to conduct artifact restoration in a unsupervised manner, which is very interesting to me. The rebuttal solve most of my concerns and I think this is a good paper for MIDL.

**Justification Of The Preliminary Rating:**

The proposed method is very interesting and useful. Conducting artifacts removal in a unsupervised manner is a valuable area to explore. But there are some unclear parts in the paper need to be addressed and the performance of artifact detection method is also a concern. Overall, this is a pretty good paper.

**Questions To Address In The Rebuttal:**

As stated in the Weaknesses and major concerns in the Detailed Comments.

**Special Issue:**

Yes

---

> ### Author Response · Authors · 2024-03-08
> **Answers to ukLv**
>
> # Weakness 1 and 1. Major concern, artifact detection
> We agree that not all methods from natural images translate well to medical images. However, we trained 10 anomaly detection methods only on pathology images and evaluated them in Table 4. We further evaluate the three best extensively in Figure 3. We think the results show that these methods can definitely work, and we do not find any anomaly detection method that was specifically developed for histopathology that includes techniques that are not already present in more generalizable methods evaluated by us. Nonetheless, we believe a more tailored approach for pathology could definitely improve upon them in the future. Regarding approaches like HistoQC, they often require a massive amount of manual labor to select the features that can detect the artifacts. Further, approaches that train with supervision on the artifacts will fall short for unobserved artifacts like the AR-cycleGAN in the restoration case, which only had information on folds and overlaps, which we outperform severely.
> # 2. Major concern, artifact detection
> We are thankful for your concern about the generation of artifacts into images without any artifacts. Our short answer is: No, they are usually not even detected to contain an artifact. HARP actually has four layers of security built into it to prevent such situations. First, we calibrate so that only 5 % of images from the validation set would get into the pipeline, in that 5 % are likely also images that contain natural artifacts. Hence, it is unlikely that an image could be inpainted without actually having an artifact. Second, we did not observe a case where an image got an artifact inpainted was selected as the final result, as our approach is conservative about selecting which inpainted image would be selected – hence the 5 artifact localization proposals. Thirdly, even when there would be such an image, it is very unlikely that an artifact would be too introduced to that image as the diffusion model was trained to restore artifact free images. Even when our diffusion model would inpaint an artifact, a newly introduced artifact would be ranked much lower by the final anomaly detection than others that reproduced tissue close to the original image -- we would only select the image considered to be the most natural. Lastly, we would only display inpainted areas to pathologists with an indication that the region was generated by an AI, as required by laws like the EU AI Act. Having built in so many layers of security, we are very confident that it would not introduce new artifacts into images that had none to begin with. However, when there are already artifacts we could observe images that were inpainted with additional artifacts in some of the 5 localization maps proposals, but those would be ranked much lower and not selected as the final image. For these reasons, we think it would not affect the downstream task, which shows in our downstream experiment.
> # Weakness 2, 1. + 2. Major concerns, artifact localization
> We use the same diffusion model for inpainting, to reconstruct complete images. If an image contains an artifact, the part with the artifact is expected to have poor reconstruction, as it deviates from the learned pattern. Further, we observed this for every instance of diffusion model we trained in the process of writing this paper, so we are fairly confident that it works as long as the artifacts are not significantly represented in the training dataset.
>
> Now, the BCSS dataset we trained on is a very clean dataset, even though it has some artifacts – mostly blood related artifacts — but they are minimal, so the model is less likely to know how to reproduce this. We only notice a slight alleviation for the blood cells in the activation map, which likely also causes them to be among the hardest artifacts to restore. However, this could be improved in two ways:
> - Improved manual cleaning of the dataset.
> - Active unlearning of artifact features.
>
> To the best of our knowledge, both methods would introduce some element of supervision, which we want to avoid as annotation costs are high or would make our model not an unsupervised approach, which we see as important to generalize to unseen artifacts.
> # Weakness 2 and 1. Minor concern
> We prompt SAM with 32x32 grid points on the entire image. We will make the entire code accessible on GitHub upon acceptance in order to ensure reproducibility. Disjointed artifacts will sometimes be segmented separately, as it is the case for scattered blood cells. For cases like this, the image could go through the pipeline multiple times, until the final artifact detection is below the anomaly threshold.
> # 2. Minor concern:
> Thank you for noticing, and we updated the PDF accordingly. It is the internal anomaly score of each method that gets bound to [0...1] by Anomalib, where 1 is a perfect anomaly and 0 is perfectly represented as in distribution.
>
> Thank you very much for your review.

---

### Meta-Review · Area_Chair_JWTb · 2024-04-02

**Recommendation:** Accept (Poster)
**Confidence:** 4

**Metareview:**

The paper describes an effective pipeline for artifact removal in histopathology images. The reviewers see merit in the work, but also raise questions regarding the methodological novelty and the practical application of the work. The authors have actively engaged during the rebuttal period, addressed some of the reviewers' concerns and adapted their paper accordingly.

---

### Decision · Program_Chairs · 2024-04-06

Accept (Poster)